# Tracking exceptional points above the lasing threshold

Kaiwen Ji [1], Qi Zhong[2], Li Ge [3,4], Gregoire Beaudoin[1], Isabelle Sagnes [1], Fabrice Raineri[1], Ramy El-Ganainy [2,5] ✉ & Alejandro M. Yacomotti[1,6] ✉

Recent studies on exceptional points (EPs) in non-Hermitian optical systems have revealed unique traits, including unidirectional invisibility, chiral mode switching and laser self-termination. In systems featuring gain/loss components, EPs are commonly accessed below the lasing threshold, i.e., in the linear regime. In this work, we experimentally demonstrate that EP singularities in coupled semiconductor nanolasers can be accessed above the lasing threshold, where they become branch points of a nonlinear dynamical system. Contrary to the common belief that unavoidable cavity detuning impedes the formation of EPs, here we demonstrate that such detuning is necessary for compensating the carrier-induced frequency shift, hence restoring the EP. Furthermore, we find that the pump imbalance at lasing EPs varies with the total pump power, enabling their continuous tracking. This work uncovers the unstable nature of EPs above laser threshold in coupled semiconductor lasers, offering promising opportunities for the realization of self-pulsing nanolaser devices and frequency combs.

Exceptional points (EPs) are algebraic branch points associated with multi-valued complex functions. In physics, EPs are associated with the spectra of non-Hermitian systems. Despite the early theoretical studies on EPs[1,2] and experimental efforts to demonstrate some of their features using microwave setups[3,4], it was not until the seminal work on parity-time (PT) symmetric potentials in quantum mechanics[5,6] and its introduction to optics[7–10] that the notion of EPs in physics has attracted considerable attention, in large part due its potential applications in optics and photonics. For recent reviews, see refs. 11–15.

Among the variety of optical platforms where EPs and their ramifications can be investigated, laser systems are particularly interesting due to the flexibility in engineering their non-Hermiticity (by adding gain and loss at will) and the ability to control their non-linearities (by choosing the appropriate material and adjusting the pump levels). This unique combination of features, coupled with the well-developed experimental techniques for measuring laser characteristics and applying different feedback schemes to control their operation, have enabled researchers to use various laser setups as a test bed for exploring a number of intriguing physical effects such as wave chaos[16], Anderson localization of light[17] and symmetry breaking[18,19].

In recent years, several experimental studies have demonstrated how PT symmetry and EPs can be utilized to control the lasing modes in multimode laser arrangements[20–24]. Subsequent theoretical works have elaborated more on the nonlinear dynamics of these systems[25–29]. An interesting feature associated with the presence of EPs in laser systems is that of laser self-termination, where applying a spatially inhomogeneous pump to a lasing device can shut down the laser action altogether[30,31]. Conversely, applying an inhomogeneous loss to a non-lasing device can lead to lasing[32]. In addition, phonon lasers based on non-Hermitian symmetries and EPs were also investigated[33–36]. In almost all the aforementioned work, the emphasis was on approaching

[1]Centre de Nanosciences et de Nanotechnologies, CNRS, Université Paris-Saclay, 10 Boulevard Thomas Gobert, 91120 Palaiseau, France. [2]Department of Physics, Michigan Technological University, Houghton, Michigan 49931, USA. [3]Department of Physics and Astronomy, College of Staten Island, CUNY, Staten Island, New York 10314, USA. [4]Graduate Center, CUNY, New York, New York 10016, USA. [5]Henes Center for Quantum Phenomena, Michigan Technological University, Houghton, Michigan 49931, USA. [6]LP2N, Institut d'Optique Graduate School, CNRS, Université de Bordeaux, 33400 Talence, France. ✉e-mail: ganainy@mtu.edu; alejandro.giacomotti@institutoptique.fr

EPs below the lasing threshold. In fact, it was explicitly demonstrated in ref. 37 that laser self-termination (or loss-induced lasing) can take place only under that condition.

EPs above laser threshold have recently been investigated theoretically, predicting that the EP laser can be stable for a large enough inversion population relaxation rate[38]. From the experimental point of view, however, even in the more recent works on PT symmetric laser[27,39,40], the relation between the lasing characteristics and the relative position of the EP with respect to the lasing threshold was not studied. While sensing devices based on a laser operating at a third-order EP were presented in ref. 41, and the signature of crossing an EP above the lasing threshold was reported in ref. 31, these systems were only analyzed within the linear coupled mode equations, which cannot capture the inherent nonlinear dynamics around EPs as a result of complex mode bifurcation and stability. In addition, the interplay between the nonlinear frequency shift induced by the amplitude-phase coupling in semiconductor lasers and the onset of EPs has received very little, if any, attention. In fact, it was concluded in ref. 42 that this nonlinear frequency shift, together with the unavoidable cavity detuning due to fabrication errors, will impede the formation of EPs in the lasing regime because of the narrow linewidths in play, and thus EPs can only be closely approached below the laser threshold.

In this work, we report on the observation of EPs above the laser threshold. Importantly, we emphasize that the EP we consider here is a self-consistent solution of the full nonlinear laser equations. It is reached when the Hamiltonian matrix governing the evolution of the electric fields becomes defective[43]. Note that this definition is different from previous work that considered linear EPs of the Jacobian matrix associated with the linearized system around arbitrary solutions[44]. In addition, we investigate the conditions for their existence and how they impact the lasing characteristics of semiconductor cavities. In particular, we study a photonic molecule laser made of two coupled

photonic crystal nanocavities and characterize the emission as the applied differential gain as well as the total power are varied. Our main finding is that the carrier-induced frequency shift breaks the effective PT symmetry of the system and thus removes the EP; however, the EP singularity can be restored by introducing an opposite frequency detuning at the fabrication stage (see Fig. 1). Furthermore, above the lasing threshold, the location of EP in terms of pump imbalance depends on the intracavity intensity, hence on the total pump power. This last feature is very different from the usual scenario where EPs are approached below the lasing threshold; in such a case, the system becomes linear and the pump imbalance at the EP is pinned down by the value of the coupling between the two cavities. As we will show shortly, this distinction between the behavior of the EP below and above the lasing threshold will be instrumental in characterizing the system under investigation. Our work provides more insight into the interplay between EPs and nonlinear interactions in coupled semiconductor nanolaser systems under nonuniform pumping. As such, it serves as a first step toward investigating more complex laser networks under extreme non-Hermitian and nonlinear conditions, such as those involving random lasers with a large number of modes[45–47].

## Results

### EPs in coupled semiconductor lasers

Figure 2 depicts a schematic of the photonic molecule laser under consideration. It consists of two coupled photonic crystal cavities implemented on an InP-based standing membrane with embedded quantum wells. This platform has been used recently for investigating physical effects such as spontaneous symmetry breaking[18,48], super-thermal light generation[49], and mesoscopic limit cycles[50]. In principle, the resonant frequency of each cavity and its coupling coefficient can be controlled by carefully engineering the cavity area, the separation between the two cavities and the size of the nanoholes. In our design, however, we only tune the resonant frequency of cavity 1 and the

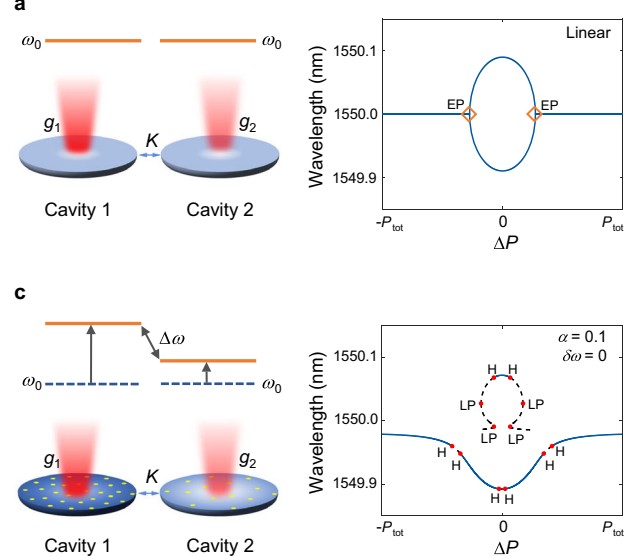

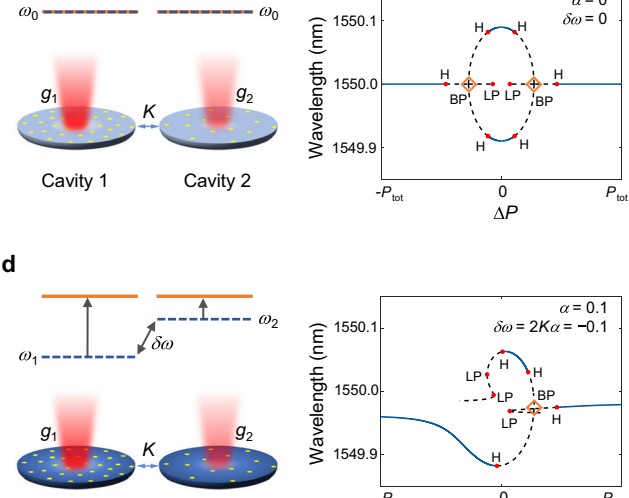

**Fig. 1 | Concept of lasing EP.** For reference, a linear PT symmetric coupled-cavity system is presented in (**a**). The case of coupled identical laser cavities with an ideal gain medium is plotted in (**b**): here, nonlinear saturation effects are taken into account but the population inversion does not induce any refractive index change, therefore there is no pump-induced frequency shift of the resonant modes ($\alpha = 0$ in Eqs. (1)). Note that the EP becomes a nonlinear (pitchfork) bifurcation point (marked by the label BP). **c** A nonzero phase-amplitude coupling in the semiconductor cavity (nonzero $\alpha$-factor) induces an asymmetric blueshift, with a net detuning $\Delta\omega$ between the two cavities and thus breaks their parity symmetry. As a result, it impedes the formation of the EP. The effective PT symmetry

($g_1 = -g_2$, $g_j = -\kappa + (n_j - n_0)\beta\gamma_\parallel/2, j = 1, 2$), and hence the formation of an EP can be restored if the (active) blueshift is compensating for by using a static redshift (i.e., introduced in the design from the beginning) $\delta\omega$ as shown in (**d**). The colors of the disks represent the sum of the frequency shift due to the carriers, $\Delta\omega_{1,2} = \alpha(n_{1,2} - n_0)\beta\gamma_\parallel/2$, and the fabrication. Here we take $P_{tot} = 3P_0$, where $P_0$ is the threshold of a single cavity. In the bifurcation diagrams, H represents the Hopf bifurcation, LP is a limit point or fold, where two steady-state solutions annihilate each other (note that the slope of the curve at these points is infinite), and BP refers to a branch point bifurcation.

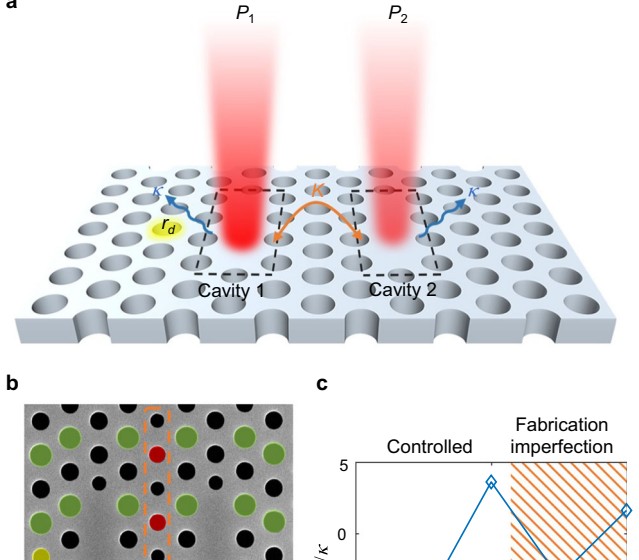

**Fig. 2 | Design and fabrication of coupled nanolasers. a** A schematic of the photonic molecule laser under investigation. It consists of two coupled photonic crystal nanocavities. The coupling strength between the two cavities can be controlled by changing the radii of the nanoholes that lie exactly at the center between the two cavities. This changes the resonant frequencies as well but by the same amount in both cavities. In turn, the individual cavity frequencies of the cavities are independently tuned by changing the size of a neighboring hole (highlighted in yellow close to cavity 1). Finally, two pump beams with different intensities providing pump rates $P_{1,2}$ are used to provide unequal gain for the two nanocavities. **b** SEM image of the fabricated sample: the material is InP with embedded InGaAsP quantum wells. The lattice constant $a = 430$nm, $r_0 = 0.266a$, $h = -0.25$ and $d = -0.1$. The coupling is controlled by the barrier between two cavities, which is displayed in an orange dashed box ($r_b = (1 + h)r_0$). The yellow hole is the detuning hole ($r_d = (1 + d)r_0$). The brown boxes indicate engineered holes to improve the beaming quality of the radiated photons ($r_{\text{beaming}} = r_0 + 0.05a$). The overlaps between the beaming holes and the barrier, colored in red, have radii of $r' = r_{\text{beaming}}(1 + h)$. **c** Detuning as a function of radii $d$. Fabrication imperfections are dominant for $d > 0$; therefore, we restrict our studies to the range of $-0.2 \leq d \leq 0$, where the cavity detuning is well-controlled by design.

coupling coefficient (detailed discussion on the design parameters will be presented later).

The lasing action of the above system can be well-described by the following rate equation model that accounts for both the field and carrier dynamics:

$$\frac{da_{1,2}}{dt} = \left[ i\omega_{1,2} - \kappa + \frac{1 + i\alpha}{2} \beta\gamma_{\parallel}(n_{1,2} - n_0) \right] a_{1,2}$$
$$+ iKa_{2,1} + F_{1,2}(t), \tag{1a}$$

$$\frac{dn_{1,2}}{dt} = P_{1,2} - \gamma_{\text{tot}} n_{1,2} - \beta\gamma_{\parallel}(n_{1,2} - n_0)|a_{1,2}|^2. \tag{1b}$$

The various variables and parameters in Eq. (1) are listed in Table 1.

As indicated above, the two cavities have identical loss coefficients. The pump imbalance $\Delta P$, defined as $\Delta P = P_1 - P_2$, is the non-Hermitian control parameter. In the absence of noise, Eq. (1) always admits a zero-field solution (with finite carrier densities but $a_{1,2} = 0$) which becomes unstable above the lasing threshold where lasing

**Table 1 | List of the variables and parameters used in Eq. (1)**

| Symbol | Physical quantity |
|---|---|
| $a_j$ | Complex fields in cavity $j$ |
| $\omega_j$ | Resonant frequency of cavity $j$ |
| $K$ | Coupling coefficient between the two cavities |
| $\kappa$ | Cavity loss rate |
| $n_O$ | Carrier number at transparency |
| $n_j$ | Carrier number in cavity $j$ |
| $\alpha$ | Phase-amplitude coupling (also known as linewidth enhancement) factor |
| $\beta$ | Spontaneous emission coefficient |
| $\gamma_{\parallel}$ | Two-level radiative recombination rate |
| $\gamma_{\text{tot}}$ | Total carrier recombination rate |
| $F_j(t)$ | Langevin noises in cavity $j$ |
| $P_j(t)$ | Pump rate in cavity $j$ |

solutions ($a_{1,2} \neq 0$) emerge. In particular, single-mode steady-state lasing solutions are defined as monochromatic solutions, i.e., $a_{1,2} = a_{1,2}(t = 0)e^{i\Omega t}$ with real $\Omega$'s[45,51,52].

The Hamiltonian used here to define the EP above the laser threshold follows from expressing Eq. (1a) in the form $-i\frac{d}{dt}\begin{bmatrix} a_1 \\ a_2 \end{bmatrix} = H_{\text{NL}}\begin{bmatrix} a_1 \\ a_2 \end{bmatrix}$. Light intensities appearing in $H_{\text{NL}}$ are evaluated at a lasing solution (see Supplementary Note 1), and the resulting Hamiltonian (i.e., the "nonlinearity-frozen" Hamiltonian) gives rise to an EP above threshold when it becomes defective. Note that since we consider here a real-valued coupling $K$, $H_{\text{NL}}$ must have an effective PT symmetry at such an EP.

Before we present the experimental results, it is instructive to first plot the lasing frequencies obtained from Eq. (1) for different values of the parameter $\alpha$. As a reference, we first show in Fig. 1a the standard EP bifurcation associated with the eigenvalues of the linear problem, i.e., with $\omega_1 = \omega_2$ and in the absence of gain and loss saturation nonlinearities (terms proportional to $|a|^2$ in Eq. (1b) neglected) and carrier-induced frequency shift ($\alpha = 0$). When the gain/loss saturation nonlinearity is included but we still take $\alpha = 0$, we observe that the eigenfrequency branching now takes place across a pitchfork bifurcation as shown in Fig. 1b. The dashed/solid lines indicate unstable/stable lasing modes, i.e., they represent unstable/stable steady-state solutions. It is straightforward to verify that the nonlinear Hamiltonian at this bifurcation point is defective, indicating that the bifurcation coincides with a nonlinear EP. Note that this coincidence is not a general behavior in nonlinear systems[43]. We also note that each branch here (including the horizontal ones) represents one lasing solution, while the horizontal branches in Fig. 1a correspond to the identical real parts of the two eigenvalues of the linear Hamiltonian.

On the other hand, when the value of $\alpha$ is finite while we still assume $\omega_1 = \omega_2$, the nonlinear EP disappears: $H_{\text{NL}}$ does not become defective along any lasing solution shown in Fig. 1c, where $\alpha = 0.1$. This can be easily explained by the carrier-induced blueshift that breaks the PT symmetry of the system. If, however, we introduce a linear frequency detuning $\delta\omega \equiv (\omega_1 - \omega_2)|_{\text{ext}}$—the subscript here indicates that this frequency detuning is introduced by external means, for instance, in the design parameters—that compensates for the nonlinear frequency shift, the EP can be restored as shown in Fig. 1d, i.e., $H_{\text{NL}}$ becomes defective. Here again, we find that the nonlinear EP coincides with the nonlinear bifurcation point. We also point out that, in some regions around the bifurcation points, all the modes are unstable. As a matter of fact, time domain integration of Eq. (1) shows that the laser output along these unstable branches is oscillatory as opposed to stable steady states (see Supplementary Note 5).

As we show in Supplementary Note 1 (Supplementary Fig. S1), even for realistic values of $\alpha = 2 - 5$, compensating for the nonlinear frequency shift through a linear frequency detuning is still possible. In fact, the carrier-induced blueshift is given by $\Delta\omega = \alpha\beta\gamma_\parallel(n_1 - n_2)/2$ and the linear frequency detuning required to compensate for this value is simply $\delta\omega = -\Delta\omega$. Assuming $n_1 > n_2$ without loss of generality, at the onset of the EP, the magnitude of the required linear shift is given by $\delta\omega = -2K\alpha$; note that, in this case, the sign indicates that cavity 1 must be externally red-detuned to compensate for the carrier-induced blueshift (see Supplementary Note 1 for detailed derivation of the above formulas). Importantly, one can also show that, above the lasing threshold, the gain difference between the cavities at the EP, $\Delta g_{12}|_{EP}$, is a function not only of the pump imbalance but also of the total pumping rate $P_{tot}$, $\Delta g_{12}|_{EP} = 2\kappa\Delta P|_{EP}/(P_{tot} - 2n_0\gamma_{tot})$ (see Supplementary Note 2). Considering that the intracavity intensity is a linear function of $P_{tot}$, we can interpret the scaling with the inverse of the total pump power as a consequence of gain saturation.

Consequently, increasing $P_{tot}$, the gain difference saturates and the pump imbalance needs to be larger to reach the PT symmetry-breaking condition at the EP, $\Delta g_{12}|_{EP} = 2K$. In the high pumping limit, $\lim_{P_{tot}\to\infty}\Delta P|_{EP}/P_{tot} = K/\kappa$. Hence, in order to observe an EP under arbitrary pump conditions, the nanocavities must satisfy the weak intercavity coupling condition ($K < \kappa$), which is an interesting outcome of our nonlinear analysis with no counterpart in linear systems.

## Coupled photonic crystal nanolasers

In order to demonstrate the very different nature of EPs above the lasing threshold, we have fabricated a photonic crystal molecule similar to that shown in Fig. 2, in which the cavity-to-cavity detuning is varied by means of the size of a side hole (yellow hole close to the left cavity). The pump profile in our experiment is controlled by using a spatial light modulator and the laser output is directed to a spectrometer to measure the lasing frequency (Fig. 3a, b).

The Q-factor for the dimer is $Q \approx 4200$ ($\kappa \approx 0.14$ THz). For a typical value of $\alpha = 3$, the condition $\delta\omega \sim -2K\alpha$ can be achieved with small coupling $K \lesssim \kappa$, for instance, $K = 0.13$ THz, and linear frequency shift of $\delta\omega = -0.74$ THz. To characterize the sample, we followed the procedure described in ref. 18 (see also "Methods"). Note that, in this case, the yellow hole in Fig. 2a is smaller than the background holes ($d = -0.1$), red-shifting cavity 1. The EP can be experimentally accessed above the lasing threshold provided: i) the total pump power exceeds twice the single laser threshold $P_0$ ($P_{tot} > 2P_0$, see Supplementary Note 3), ii) introducing a positive pump imbalance $\Delta P$ between the two cavities that blue shifts cavity 1 with respect to cavity 2 so as to compensate for the external red-detuning. The non-Hermitian parameter $\Delta P$ can be continuously varied so as to approach $\Delta P|_{EP}$ (see detailed discussion in Supplementary Note 2).

The left panels of Fig. 3c−f depict the experimental results characterizing the emission wavelength as a function of $\Delta P$ for different values of the total pump power. All the different cases share some

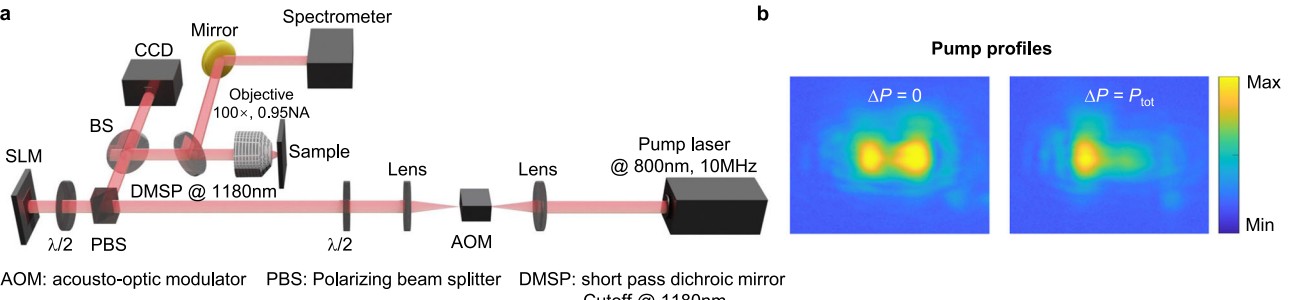

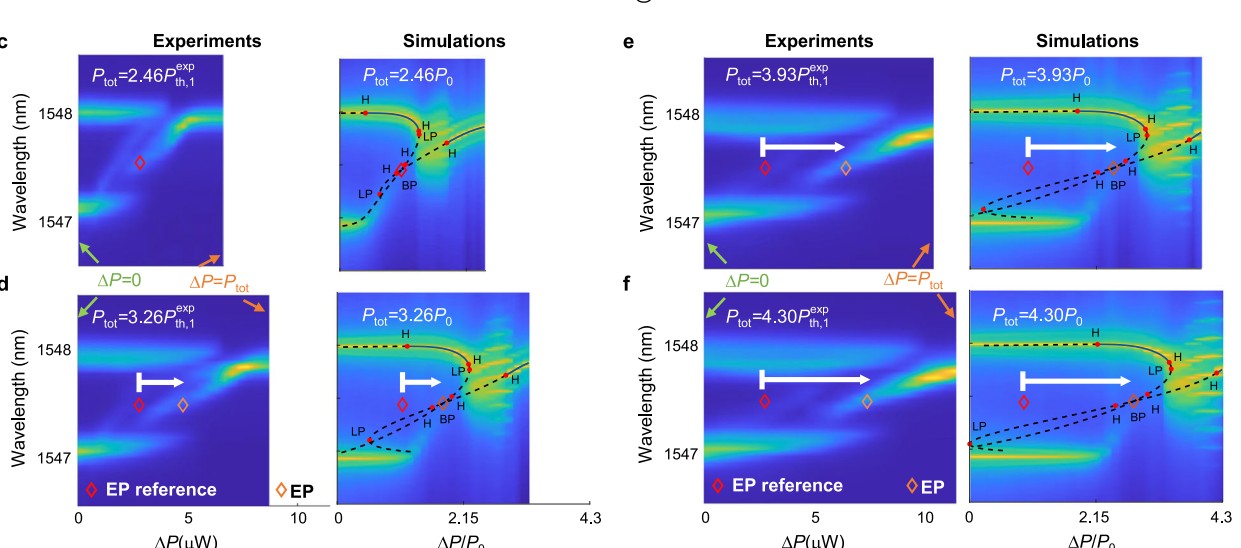

**Fig. 3 | Tracking EPs above the lasing threshold. a** A schematic of the experimental setup used in this work, where an SLM is used to engineer the pump profile. **b** Examples of two pump patterns corresponding to the extreme cases of $\Delta P = 0$ (both cavities are equally pumped) and $\Delta P = P_{tot}$ (only cavity 1 is pumped). **c−f** Experimental (left panels) and simulation (right panels) results for the lasing modes, characterized by their emission wavelength versus the pump imbalance $\Delta P$ for different total pump values $P_{tot}$. The diamonds represent the location of the EP.

For reference, the EP from $P_{tot} = 2.46P_{th,1}^{exp}$, where $P_{th,1}^{exp} \approx 2.67\mu$W is the threshold when a single cavity 1 is pumped, is also indicated in all panels as a red diamond; the shift of the EP toward higher positive values of $\Delta P$ as $P_{tot}$ is increased can be clearly observed. Here the measured coupling and detuning are $K/\kappa = 0.95$ and $\delta\omega/\kappa = -5.25$, respectively. For details on how the measurements were performed, see "Methods". In the simulation, the detuning is chosen to be $\delta\omega/\kappa = 2\alpha K = -5.70$.

generic qualitative behavior. First, for $\Delta P = 0$, there are two distinct spectral peaks indicating that the two modes of the photonic dimer are participating in the lasing action. This feature is consistent with the bifurcation diagrams (Fig. 3c–f, right panels, where dashed lines account for unstable steady states), which predict that both modes are unstable for $\Delta P = 0$ and give rise to mode beating limit cycles[50] (Supplementary Fig. S4b, bottom panel, in Supplementary Note 5). The measured frequency splitting between the two modes is approximately 0.79 THz. This compares well with the linear model, which predicts the presence of two supermodes oscillating at different frequencies with a splitting given by $\Delta\Omega = 2\sqrt{(\delta\omega/2)^2 + K^2} \sim 0.785$ THz. As $\Delta P$ is increased, multimode features appear in the laser spectra, which we relate to additional instabilities. Importantly, these are predicted by the model close to EP singularities, meaning that they can only be observed under detuning compensation conditions, and therefore they can be taken as a signature of the proximity to the EP. These instabilities generate side-bands in the theoretical spectra, for instance, the observed frequency combs in the simulation panels of Fig. 3d–f, at the right side of EPs where no stable states exist. The predicted comb frequency is $\Delta\nu_{\mathrm{comb}} \sim 12$ GHz, which is governed by the slow electronic lifetime of the system, and it can be related to Q-switch-like pulsation in coupled nanolasers[53]. The predicted self-pulsing arises from a Hopf bifurcation (the right-most H-dots in the simulation panels), and the corresponding time trace is shown in Supplementary Fig. S4b (top) of the Supplementary Information. Such spectral features cannot be experimentally resolved by our setup since: (1) the spectrometer resolution is ~30 GHz, and (2) the pulsed pumping scheme (100-ps pulse duration) is not well adapted to observe such a 12-GHz pulsation. Finally, above a certain threshold for $\Delta P$, the multimode emission collapses and only one lasing mode is measured. The observed mode-structure is in good quantitative agreement with the numerical solutions of Eq. (1) with added noise terms (color maps in the right panels of Fig. 3c–f). The details of how these solutions are obtained numerically are discussed in Supplementary Note 5 and 6. Another interesting generic observation from Fig. 3 is that, as $\Delta P$ increases, the lasing emission in the PT-broken-like phase becomes red-shifted (e.g., Fig. 3e, branch crossing the EP, from $\Delta P \sim 5\mu W$ to $10\mu W$). This is counter-intuitive since in this case, the applied pump to cavity 1 is increased. One thus may expect a blueshift due to the amplitude-phase coupling (the $\alpha$ parameter). A close inspection, however, reveals that in this PT-broken-like phase, the lasing threshold is lower than in the PT-unbroken-like phase. The gain clamping will thus result in smaller carrier density and consequently a redshift from the operation in the PT-unbroken-like phase as well as at the EP (see Supplementary Note 3 for a detailed discussion).

In addition, contrary to what one would expect from using a simplified linear model, a close inspection of the experimental and theoretical data presented in Fig. 3c–f reveals a rather interesting trend, namely that the location of this EP shifts to higher values of $\Delta P$ as the total pump power is increased. In this work, this feature is taken as strong evidence supporting the presence of an EP above the lasing threshold. As a matter of fact, it can be easily shown (see Supplementary Note 2) that $\Delta P$ varies as a function of both the coupling $K$ and the total pump $P_{\mathrm{tot}}$ in the following way:

$$\Delta P|_{\mathrm{EP}} = \frac{K(P_{\mathrm{tot}} - 2n_0\gamma_{\mathrm{tot}})}{\kappa}. \qquad (2)$$

Three distinct regimes can be identified based on the value of $K/\kappa$: (1) $K/\kappa < 1$, (2) $1 < K/\kappa < 1 + \frac{1}{2\kappa}n_0\beta\gamma_{\parallel}$, and (3) $1 + \frac{1}{2\kappa}n_0\beta\gamma_{\parallel} < K/\kappa$. In the first of these regimes, the EP can be accessed above the lasing threshold for any value of $P_{\mathrm{tot}}$ but below the horizontal line given by $\Delta P/P_{\mathrm{tot}} < K/\kappa$ (see top panel of Fig. 4). In the second regime, the EP can be accessed only for a finite range for $P_{\mathrm{tot}}$ as shown in the lower panel of Fig. 4.

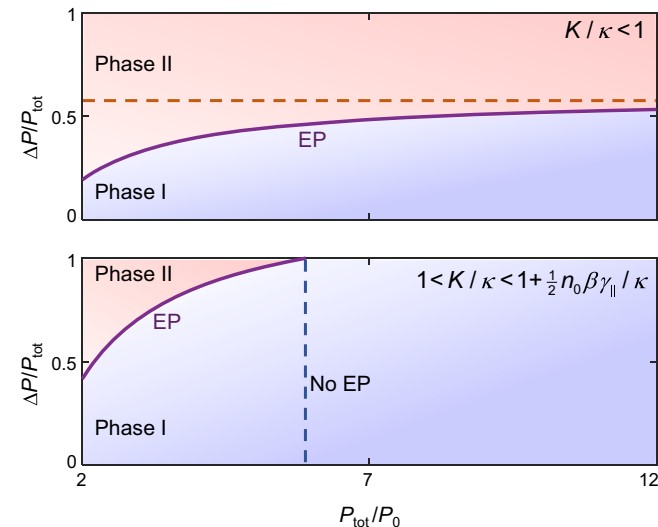

**Fig. 4 | Different operation regimes.** Top panel depicts the operating regimes when $K/\kappa < 1$. In this case, exceptional points can be accessed above the lasing threshold for any value of $\Delta P|_{\mathrm{EP}}/P_{\mathrm{tot}}$ but only for $\Delta P/P_{\mathrm{tot}} < K/\kappa$ (the asymptotic red dashed line). On the other hand, as shown in the lower panel, when $1 < K/\kappa < 1 + \frac{1}{2\kappa}n_0\beta\gamma_{\parallel}$, the EPs can be accessed only in the domain defined by the condition $P_{\mathrm{tot}} < \frac{2K n_0\gamma_{\mathrm{tot}}}{K-\kappa}$ (see vertical dashed line). Here we denote Phase I as a PT-unbroken-like bimodal phase and Phase II as a PT-broken-like single-mode phase (further details in the text). Finally, when $1 + \frac{1}{2\kappa}n_0\beta\gamma_{\parallel} < K/\kappa$, EPs cannot be accessed at all above the lasing threshold. Here we take $K/\kappa = 0.6$ (top panel) and $K/\kappa = 1.3$ (lower panel) as examples.

Finally, in the third regime, the EP cannot be accessed at all above the lasing threshold. Note that in Fig. 4, we denoted the areas below and above the exceptional line as phases I and II, respectively. In general, these phases cannot be associated with exact and broken PT phases, mainly due to the complex nature of the nonlinear bifurcation and the finite frequency shift due to the $\alpha$ factor. However, in cases where $\alpha = 0$ (such as the case in gas and rare-earth-doped solid-state lasers) or when the frequency shift due $\alpha$ is negligible compared to the coupling strength between the two resonators, one can make such a correspondence between phases I and II on one hand and the exact and broken PT phases on the other hand.

Another intriguing property of nonlinear EP is that the lasing threshold is exactly $2P_0$ (see Supplementary Note 3 for detailed calculation). To gain access to the EP both below and above the threshold, we fix the pumping rate of cavity one, $P_1$ and increase $P_2$. In the first case, we chose $P_1 = 1.1P_0$. For this choice ($P_{\mathrm{tot}} < 2P_0$), the EP occurs below the lasing threshold. As can be seen from the experimental results shown in the top panel of Fig. 5a, under this condition, the system experiences laser self-termination and revival. Numerical calculations depicted in the lower panel also confirm these results. On the other hand, when the same experiment is repeated for $P_1 = 2P_0$, the EP is accessed above the lasing threshold and self-termination/revival behavior disappears, filling the laser extinction gap. These results are in good agreement with the linear model considered in ref. 37 for analyzing laser self-termination.

Finally, we have also fabricated a second sample with a relatively weak frequency detuning of $\delta\omega/\kappa = 1.19 \ll 2K\alpha = 22.38$ between the two photonic crystal cavities. This detuning cannot compensate for the carrier-induced frequency shift. Hence, an asymmetric pump cannot be used to access an EP in this sample. As discussed before, in this case, carrier-induced frequency shift will break the parity symmetry between the two cavities, and as a result, the system will not exhibit any EP. As a matter of fact, the experimental data in this case, which we present in Supplementary Fig. S6 in Supplementary Information, reveal that the lasing characteristics are almost insensitive to $P_{\mathrm{tot}}$.

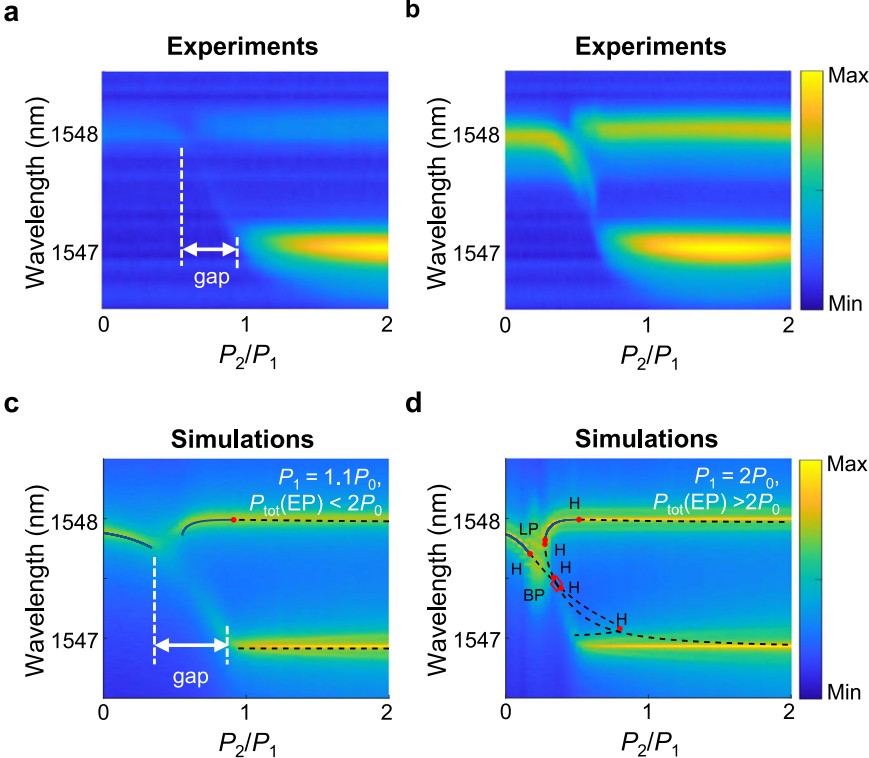

**Fig. 5 | Lasing versus non-lasing EPs.** Experimental (top row) and theoretical (lower row) results for accessing EP below and above the lasing threshold. In these figures, $P_1$ is kept constant and $P_2$ is varied. **a** $P_{tot} < 2P_0$ ensures that the EP can be accessed only below the lasing threshold, as evidenced by the self-termination that takes place as $P_2/P_1$ is increased. **b** $P_{tot} > 2P_0$ to ensure that the EP can be accessed above the lasing threshold. Here we observe mode switching without any self-termination effects. In all the figures, the color maps are presented in log scale for clarity. **c** and **d** display the simulation results, which are in good agreement with the experiments.

Importantly, the lack of compensation, hence the absence of an EP, does not impede transitions from two coexisting modes for small $|\Delta P|$ to a single localized mode for large $|\Delta P|$. While in many experimental examples such a mode transition is usually interpreted as a certain proximity to an EP, we stress the fact that such a phase transition, if any, instead of resulting from a weakly perturbed EP-bifurcation structure, takes place far from any EP, in the sense of Fig. 1c (see also Supplementary Fig. S6 in Supplementary Information).

## Discussion

While the recent interest in PT symmetry has attracted considerable attention, most of these studies have focused on utilizing EPs below the lasing threshold to engineer the cold cavity modes assuming that the character of the modes will remain intact above the lasing threshold, where the system is fundamentally linear. Even though this approach has proven useful in certain practical situations, it has two main drawbacks. First, it ignores the rich dynamics that can arise due to the interplay between nonlinearity and non-Hermitian effects associated with EPs. Second, it may fail in systems where the addition of gain alters the nature of the modes (see, for example, ref. 54). In the present work, we have bridged this gap by systematically demonstrating how EPs can be accessed and tracked above the lasing threshold in coupled semiconductor photonic crystal nanolasers. Contrary to previous studies that considered unavoidable cavity detuning as a nuisance that impedes the formation of an EP above the lasing threshold[42], here we show that controllable detuning is actually a key ingredient for compensating the carrier-induced frequency shift and hence steering the system toward PT symmetry and EPs. Notably, our analysis shows that, in the nonlinear regime above the threshold, the EP becomes a nonlinear bifurcation point and solutions around it acquire a rich dynamical behavior. Namely, in some regions around

this point, we find that all the steady-state solutions are unstable, and the lasing action becomes oscillatory. Such instabilities apply to the particular case of class-B laser systems—to which semiconductor lasers belong—(see ref. 55 and references therein), where the population inversion decay rate $\gamma_{tot}$ is much smaller than the cavity damping rate $\kappa$. In turn, most laser models usually utilized in the literature so far for describing PT and EP-related phenomena are class-A laser systems, where the atomic population is assumed fast and can therefore be adiabatically eliminated; as a result, the EPs become stable steady states[38] which, as we show in this work, is a very different scenario from what is expected for semiconductor cavities.

A particularly interesting outcome of our work is the realization of a certain constraint between the intercavity coupling and losses for the EP to form above the lasing threshold—a feature that does not have a counterpart in linear systems. While our work here focuses on two single-mode coupled cavities, it opens the door for future investigations on more complex laser systems. For instance, it is well known that interesting non-Hermitian effects can arise in deformed cavities having a large number of modes under passive conditions[56,57]. Much less is known about the non-Hermitian effects in these systems in the nonlinear lasing regime. Similarly, understanding the interplay between non-Hermitian effects and disorder has only recently started to emerge[58] but again in non-lasing setups. Extending this understanding to the nonlinear lasing regime can unlock more rich physics that so far has escaped attention. Importantly, we would like to comment on the lasing linewidth in our experiment. Even though we have not performed precise measurements of the emission linewidth, our experimental and numerical data in Fig. 3 clearly demonstrate that the linewidth is finite in the presence of an EP. This in turn confirms the breakdown of the linewidth enhancement formula given by the Petermann

factor[59–62] since the latter diverges at EPs (see also the discussion about the Petermann factor in phonon lasers[63]). At the same time, it is clear that a relatively broader linewidth due to the overlap between different lasing modes can occur close the EP, which in fact is a nonlinear bifurcation point. This could make it more difficult to discriminate between the lasing frequencies, which may degrade the operation of EP-based laser sensors. Furthermore, the instability of the steady-state solutions at the EPs may pose an additional challenge for these sensors. On the other hand, some of those instabilities in the proximity to EPs have been shown to generate Q-switch self-pulsing, which opens interesting prospects for the realization of non-steady laser sources such as self-pulsing nano-laser devices and nanophotonic frequency combs. Additionally, it will be interesting to extend our current study to multimode laser systems where the interplay between non-Hermiticity and direct modal interaction may give rise to even more interesting and complex behavior, as well as to the quantum domain[64–66]. We will investigate these open questions in future works.

## Methods

Two coupled photonic crystal cavities have been fabricated in an Indium Phosphide (InP) membrane (256 nm-thickness) with four embedded $InGa_{0.17}As_{0.76}P$ quantum wells.

The cavities are pumped using a pulsed laser ($\lambda = 800$nm, 100 ps duration and 10 MHz repetition rate) to reduce thermal effects. The global intensity of the pump is controlled by an acousto-optic modulator (AOM). To control the pump profile across the cavities in an independent manner, we use a spatial light modulator (SLM) to reshape the pump profile. The SLM is operated in amplitude mode, in which we use two $\lambda/2$ plates to maximize both intensity and the contrast, respectively [the first $\lambda/2$ plate (close to the AOM) rotates the polarization plane for maximum transmission through the polarizing beam splitter; the second $\lambda/2$ plate (close to the SLM) rotates the polarization of the incident light to 45° to achieve the higher contrast between the pump pattern and the unwanted background]. An infrared AR-coated microscope objective with ×100 magnification and N.A. = 0.95 focuses down the pump on the sample. The free-space emission is collected with the same objective and then spectrally resolved with a spectrometer.

The system parameters are measured using the same protocol as in the Supplementary information of ref. 18. In particular, the coupling and cavity-to-cavity detuning are measured as follows:

i. Pumping both cavities to obtain the split lasing modes $\omega_\pm$.
ii. Pumping cavity 1 to obtain the blue-shifted frequency of that cavity, $\omega_1$.
iii. Same for cavity 2 to obtain $\omega_2$.

The eigenvalues of the linear Hamiltonian are used subsequently,

$$\begin{aligned} \omega_+ - \omega_- &= \sqrt{4K^2 + (\omega_1 - \omega_2)^2}, \\ \omega_+ + \omega_- &= \omega_1 + \omega_2. \end{aligned}$$

To locate and track the EP experimentally, we first pump the red-detuned cavity (cavity 1) and measure the threshold, $P_{\text{th},1}^{\text{exp}}$. As we have shown in Supplementary Note 3, when the coupling is weak, one can use the approximation $P_{\text{th},1}^{\text{exp}} \approx P_0$. The EP can therefore be located approximating Eq. (2) as:

$$\frac{\Delta P|_{\text{EP}}^{\text{exp}}}{P_{\text{th},1}^{\text{exp}}} \approx \frac{K}{\kappa}\left(\frac{P_{\text{tot}}^{\text{exp}}}{P_{\text{th},1}^{\text{exp}}} - 2 + \frac{4}{2 + \beta n_0 \gamma_\parallel/\kappa}\right),$$

where the parameters are listed in Supplementary Table S1 of the Supplementary Information.

The lasing wavelength at the EP is $\lambda_{\text{EP}} = 2\pi c/\Omega_{\text{EP}}$, where $\Omega_{\text{EP}}$ is given by (see Supplementary Note 3 for details)

$$\Omega_{\text{EP}} = \omega_1 + \alpha(\kappa + K).$$

## Data availability

Data is available upon request from the corresponding author.

## Code availability

Bifurcation analysis was performed using the Matlab package Matcont, which can be accessed at https://sourceforge.net/projects/matcont/.

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

## Acknowledgements

K.J. acknowledges support from the China Scholarship Council (Grant No.202006970015). L.G. acknowledges support from the National Sci-ence Foundation under Grant No. PHY-1847240. R.E. acknowledges support from the Air Force Office of Scientific Research (FA9550-21-1-0202). This work is partially supported by the French National Research Agency (ANR), Grants No. ANR UNIQ DS078 and ANR-22-CE24-0012-01, the European Union in the form of Marie Skłodowska-Curie Action grant MSCA-841351, and by the RENATECH network.

## Author contributions

A.M.Y., R.E. and L.G. conceived the project. A.M.Y. supervised the project. K.J. performed the experiment and simulations with feedback from Q.Z., R.E., L.G. and A.M.Y. G.B. and I.S. contributed to the sample growth and F.R. to the nanofabrication. K.J., A.M.Y., Q.Z., R.E. and L.G. contributed to the manuscript preparation.

## Competing interests

The authors declare no competing interests.
