## [Peer Review File · Nature Communications]

REVIEWER COMMENTS

Reviewer #1 (Remarks to the Author):

Subject: Review of NCOMMS-23-08002-T

review of the manuscript "Tracking exceptional points above laser threshold" by Dr El-Ganainy and colleagues

The current manuscript presents an experimental investigation of the spectral features of PT symmetric photonic crystal nanolasers under various differential pump conditions. The experimental results are supported by detailed theoretical analysis. This work clarifies several important issues:

(i) It presents a clear definition of the notion of exceptional points (EPs) in the presence of nonlinearity. This is done by introducing the concept of the frozen Hamiltonian;

(ii) It investigates the role of the amplitude-phase coupling described by the Henry factor on the presence of EPs;

(iii) It demonstrates that compensating the nonlinear frequency shift by a linear detuning is necessary to observe EP bifurcation;

(iv) It shows that the interaction between nonlinearity and non-Hermiticity leads to a more complex bifurcation behavior than previously thought (e.g. in Fig. 1(b), the bifurcation is a pitchfork bifurcation rather than the usual linear EP bifurcation).

The topic of the work is timely and important, and the results are presented clearly. I can recommend this very interesting work for publication in Nature communications after the authors address the following issues:

1. In Page 2, left column, Fig. 1(d) is mentioned before the rest of the panels. Maybe the authors can rearrange the discussion in order to cite the different panels in the correct order.

2. In the figure caption of Fig.1, what are exactly the LP points. The authors mention that these are limit points. But that still needs more clarification. What are these points exactly?

3. In the same figure, the authors also denote some points as BP without definition. I am guessing they mean branching points but that needs to be spelled out explicitly.

4. In all the figures, the authors use dashed lines to denote “unstable laser solutions”. This needs more clarification. By unstable here, do they mean unstable steady state solutions or unstable limit cycles? I am guessing it is the first but again that needs to be explained explicitly.

5. When the nonlinear frequency shift is compensated with linear detuning, the EP is restored. However, in that case, the authors mention that the bifurcation becomes transcritical. This point needs more discussion. What is exactly the criterion used to make this claim?

6. At the EP transcritical bifurcation, how does the eigenfrequency branching scale as a function of ΔP ? Visually, it does not look like the familiar square root splitting associated with linear EPs. Can the authors comment on this point?

7. In all the figures, I can see that EPs coincide with nonlinear bifurcation. Is this a general rule that applies to any nonlinear system?

8. The experimental and numerical results show that the spectral lines become fuzzy and difficult to discriminate close to the EP. Is this solely due to the noise effects or is there another factor? It would be helpful if the authors comment on the resolution of the spectrometer used in taking these measurements.

9. The discussion on the Petermann factor (PF) is missing an important study that investigated PF in phonon lasers: Nature Photonics, volume 12, 479–484 (2018).

10. In Figure 2, panel (c), the variation of $\Delta \omega / \kappa$ as a function of the variation of the hole radius, d , is not monotonic in the dashed region, i.e., it decreases first before it increases again. Can the author comment on this and give a possible explanation?

In addition to the above, there other minor comments:

1. I think the title perhaps should be modified to "Tracking exceptional points above the lasing threshold"
2. Figure S6 is small, maybe the authors can enlarge it or rearrange it to be larger.
3. There are some issues in the references formatting. For example, in Ref 5, the "H" in the words Hermitian and Hamiltonian should be capped. This problem is also repeated in Refs. 12,51,55,57.
4. Similarly, in Ref, 17, it should be "Anderson".
5. In Ref 40, the publisher information is not needed.
6. The introduction could place this work in a broader context by citing additional relevant references, for example: Phys. Rev. Lett. 113, 053604, 2014; Phys. Rev. B 92, 115407, 2015; Phys. Rev. Lett. 117, 110802, 2016.

In summary, this very interesting work investigates experimentally the interplay between nonlinearity and non-Hermiticity in PT symmetric laser systems. Given the importance of understanding these effects for building new lasers and sensing devices, and the fact that these effects have been so far overlooked when dealing with PT symmetric lasers (most of the existing literature analyze these systems using only linearized models), I do find the topic important and timely. Therefore, I recommend this work for publication in Nature Communications, after the authors address the above points thoroughly and successfully.

Reviewer #2 (Remarks to the Author):

In this manuscript, Ji et al provides an experimental and theoretical study of the effect of non-Hermiticity and PT-symmetry on coupled photonic crystal nanolasers.

This study is interesting and timely because it fills a knowledge gap, that is the interplay of non-Hermitian spectral degeneracies (EPs) and nonlinearity, in particular the nonlinearity of lasers. The study of non-Hermiticity and nonlinearity in a laser system brings a more complex setting than is observable non-Hermitian linear system. Therefore, this field has remained largely unexplored. I am happy to see that these authors have taken such a difficult task and provided a very clear demonstration of how non-Hermitian spectral degeneracies, EPs, emerge in the presence of nonlinearity and how it affects the laser dynamics. It is well-known that nonlinearity may move a system away from an EP or bring it to an EP. Therefore, it is often considered as detrimental if one wants to work at an EP. Here, the authors show that if nonlinearity pushes a system from an EP, one can compensate this by introducing linear tuning. In these demonstrations, the authors use pump power as a tunable parameter to track the spectral location of EPs above laser threshold.

I think this paper deserves publication in Nat. Comm, and I recommend it without any reservation. Some minor points that the authors should address in a revised manuscript are as follows:

1. I am not familiar with the notion of limit points? What do these correspond to in a nonlinear physical system? Do these points exist in linear systems?
2. Can the authors elaborate more on the relation between EPs and Hopf bifurcations in nonlinear systems?
3. Could the authors comment on if their findings will change in the case of multimode lasers?

Reviewer #3 (Remarks to the Author):

The authors report a study of EP above lasing threshold in a system of coupled micro-resonators. The authors show that detuning of one of the micro-resonators must be designed in to counteract the frequency shift induced by the carriers in the semiconductors (InGaAsP) to achieve EP in a lasing stage. The authors show spectral plots (Fig 3) as the pump power distribution between the two micro-resonators are varied. Of note, the lasing EPs shown in this work are unstable.

While I commend the authors on their effort, the last point presents itself as a serious drawback to me. I cannot think of a use for such a laser presented in this work. I do not think that rich dynamics alone can justify its significance. Unless the authors can persuade otherwise, I regret I cannot support the publication of this work.

Response letter
Tracking exceptional points above the lasing threshold

Below is our response to the referees' comments/recommendations. All revisions in the manuscript and SM are highlighted in red.

Reviewer #1:

Overall Comment

The current manuscript presents an experimental investigation of the spectral features of PT symmetric photonic crystal nanolasers under various differential pump conditions. The experimental results are supported by detailed theoretical analysis. This work clarifies several important issues:

- (i) It presents a clear definition of the notion of exceptional points (EPs) in the presence of nonlinearity. This is done by introducing the concept of the frozen Hamiltonian;
- (ii) It investigates the role of the amplitude-phase coupling described by the Henry factor on the presence of EPs;
- (iii) It demonstrates that compensating the nonlinear frequency shift by a linear detuning is necessary to observe EP bifurcation;
- (iv) It shows that the interaction between nonlinearity and non-Hermiticity leads to a more complex bifurcation behavior than previously thought (e.g. in Fig. 1(b), the bifurcation is a pitchfork bifurcation rather than the usual linear EP bifurcation).

The topic of the work is timely and important, and the results are presented clearly. I can recommend this very interesting work for publication in Nature communications after the authors address the following issues:

Our response:

We thank the reviewer for summarizing our work and highlighting its novelty and importance, and for recommending it for publication in Nature Communications.

Comment 1

In Page 2, left column, Fig. 1(d) is mentioned before the rest of the panels. Maybe the authors can rearrange the discussion in order to cite the different panels in the correct order.

Our response:

We thank the Referee for pointing this out. We have revised the text to address this issue.

Comment 2

In the figure caption of Fig.1, what are exactly the LP points. The authors mention that these are limit points. But that still needs more clarification. What are these points exactly?

Our response:

We apologize for not having specified this before. Here, LP –as computed with bifurcation continuation packages– stands for limit point or *fold*, where two steady state solutions annihilate each other. In bifurcation theory, a typical fold is a saddle-node bifurcation, where “node” stands for stable equilibrium. In our case the folds are not saddle-nodes because solutions in a neighborhood of the EPs are always unstable, and therefore we used the generic term *fold*. We have now clarified this definition in the figure caption.

Comment 3

In the same figure, the authors also denote some points as BP without definition. I am guessing they mean branching points but that needs to be spelled out explicitly.

Our response:

The Referee is indeed correct. Here BP refers to branching or bifurcation point. We have now clarified this in the figure caption.

Comment 4

In all the figures, the authors use dashed lines to denote “unstable laser solutions”. This needs more clarification. By unstable here, do they mean unstable steady state solutions or unstable limit cycles? I am guessing it is the first but again that needs to be explained explicitly.

Our response:

The instability here refers to steady state solutions. This has been now clarified in the main text by adding the sentence: “*i.e.*, they represent unstable/stable steady state solutions.” (page 3, 1st column, 1st par.).

Comment 5

When the nonlinear frequency shift is compensated with linear detuning, the EP is restored. However, in that case, the authors mention that the bifurcation becomes transcritical. This point needs more discussion. What is exactly the criterion used to make this claim?

Our response:

We acknowledge the reviewer’s question. In pitchfork bifurcations, the number of steady state solutions switch from 1 to 3 at either side of the branch point. At a transcritical bifurcation, the number of solutions does not change (always 2 solutions at either side), only the stability of the solutions can change at the bifurcation point.

Comment 6

At the EP transcritical bifurcation, how does the eigenfrequency branching scale as a function of ΔP ? Visually, it does not look like the familiar square root splitting associated with linear EPs. Can the authors comment on this point?

Our response:

The referee raises a very interesting point. It is absolutely correct; the scaling is, in fact, linear, meaning that the frequencies change linearly with respect to ΔP . This behavior is not unique to nonlinear systems, though (See, for example, PRL 129, 243901 (2022) and PRB 107, 104106 (2023)). Nevertheless, if some other parameter is varied, for instance the detuning, the model indeed predicts square root scaling. The dependence on $\delta\omega$ has been added as Figs. S1c and S1d in the SM. We also add the following sentence in S1 of the SM (page 1, 2nd column, last par.): “Unlike a pitchfork, the bifurcation at the EP as ΔP is changed is a *transcritical* one: locally, there are two fixed points before and after the branch point. Importantly, while the frequency scaling is linear in the vicinity of the EP, it becomes square-root as the bifurcation parameter is the detuning [Figs. S1(c) and (d)]”.

Comment 7

In all the figures, I can see that EPs coincide with nonlinear bifurcation. Is this a general rule that applies to any nonlinear system?

Our response:

This is indeed a good question. It has been studied in a previous work by one of the coauthors [Phys. Rev. A 94, 013837 (2016)], where it was shown that, in general, EPs do not have to coincide with the nonlinear bifurcation. Note that this point is explicitly mentioned in page 3, 1st column, 1st paragraph: “Note that this coincidence is not a general behavior in nonlinear systems [43]”.

Comment 8

The experimental and numerical results show that the spectral lines become fuzzy and difficult to discriminate close to the EP. Is this solely due to the noise effects or is there another factor? It would be helpful if the authors comment on the resolution of the spectrometer used in taking these measurements.

Our response:

We thank the referee for raising this important point. We believe that the fuzzy lines are actually related to non-stationary dynamics close to EPs, enhanced by the presence of noise, as predicted by the model. For instance, in the numerical maps of Figs. 3d, 3e and 3f, we clearly observe spectrally narrow side-bands (a frequency comb) in the proximity to the EPs, namely at the right-hand side of the EP where no stable steady states exist. The predicted comb frequency is ~ 12 GHz (corresponding to $\Delta\lambda \sim 0.1$ nm) which cannot be resolved by our spectrometer, whose resolution is ~ 200 pm (~ 30 GHz). In addition, because of thermal management, the optical pump are 100-ps pulses, which do not reach cw conditions to reveal such instabilities. We added the following comment in the text (p. 5, 1st column, 1st par), and a new Ref. [51] that already predicted such Q-switch-like pulsation in coupled nanolasers: “These instabilities generate side-bands in the numerical spectra, for instance the observed frequency combs in the simulation panels of Figs. 3(d)-(f), at the right-hand side of EPs where no stable states exist. The predicted comb frequency is $\Delta\nu_{\text{comb}} \sim 12$ GHz, which is governed by the slow electronic lifetime of the system and it can be related to Q-switch like pulsation in coupled nanolasers [51]. Such spectral features cannot be experimentally resolved by in our setup since: i) the spectrometer resolution is ~ 30 GHz, and ii) the pulsed pumping scheme (100-ps pulse duration) is not well adapted to observe such a 12 GHz pulsation.”

Comment 9

The discussion on the Petermann factor (PF) is missing an important study that investigated PF in phonon lasers: Nature Photonics, volume 12, 479–484 (2018).

Our response:

We thank the Referee for bringing this important work to our attention. We have now cited it (Ref. [65]) properly.

Comment 10

In Figure 2, panel (c), the variation of $\delta\omega/\kappa$ as a function of the variation of the hole radius, d , is not monotonic in the dashed region, i.e., it decreases first before it increases again. Can the author comment on this and give a possible explanation?

Our response:

As we explained in the text this comes from fabrication imperfections. The yellow hole is intended to shift the left-cavity frequency, since its radius is modified (here decreased) by ~ 10 nm, leading to a frequency detuning of ~ 5 half-linewidths. The fabrication tolerance, on the other hand, is about a 1-2 linewidths RMS. However, this can fluctuate well beyond the RMS from sample to sample. This is the best structure quality we can fabricate, and we do not have control on this, unfortunately. Here, the two samples with $d > 0$ seem to be seriously affected by fabrication-imperfection-induced detuning, eventually to overcoming the deterministic one.

Comment 11

I think the title perhaps should be modified to “Tracking exceptional points above the lasing threshold”

Our response:

We thank the Referee for the suggestion. We have now revised the title.

Comment 12

Figure S6 is small, maybe the authors can enlarge it or rearrange it to be larger.

Our response:

Following the Referee’s suggestion, we have enlarged the figure for better visualization.

Comment 13

- There are some issues in the references formatting. For example, in Ref 5, the “H” in the words Hermitian and Hamiltonian should be capped. This problem is also repeated in Refs. 12,51,55,57.
- Similarly, in Ref, 17, it should be “Anderson”.
- In Ref 40, the publisher information is not needed.

Our response:

We thank the Referee for pointing out these problems with the reference list. We have now fixed them.

Comment 14

The introduction could place this work in a broader context by citing additional relevant references, for example: Phys. Rev. Lett. 113, 053604, 2014; Phys. Rev. B 92, 115407, 2015; Phys. Rev. Lett. 117, 110802, 2016.

Our response:

We thank the Referee for the useful suggestion. We have now included these references (Refs. [33,34]), as well as others, in the introduction.

Comment 15

In summary, this very interesting work investigates experimentally the interplay between nonlinearity and non-Hermiticity in PT symmetric laser systems. Given the importance of understanding these effects for building new lasers and sensing devices, and the fact that these effects have been so far overlooked when dealing with PT symmetric lasers (most of the existing literature analyze these systems using only linearized models), I do find the topic important and timely. Therefore, I recommend this work for publication in Nature Communications, after the authors address the above points thoroughly and successfully.

Our response:

We thank the Referee again for the very useful suggestions, and for his/her positive assessment of our work and recommending it for publication in Nature Communications.

Reviewer #2:

Comment 1

In this manuscript, Ji et al provides an experimental and theoretical study of the effect of non-Hermiticity and PT-symmetry on coupled photonic crystal nanolasers.

This study is interesting and timely because it fills a knowledge gap, that is the interplay of non-Hermitian spectral degeneracies (EPs) and nonlinearity, in particular the nonlinearity of lasers. The study of non-Hermiticity and nonlinearity in a laser system brings a more complex setting than is observable non-Hermitian linear system. Therefore, this field has remained largely unexplored. I am happy to see that these authors have taken such a difficult task and provided a very clear demonstration of how non-Hermitian spectral degeneracies, EPs, emerge in the presence of nonlinearity and how it affects the laser dynamics. It is well-known that nonlinearity may move a system away from an EP or bring it to an EP. Therefore, it is open considered as detrimental if one wants to work at an EP. Here, the authors show that if nonlinearity pushes a system from an EP, one can compensate this by introducing linear tuning. In these demonstrations, the authors use pump power as a tunable parameter to track the spectral location of EPs above laser threshold.

I think this paper deserves publication in Nat. Comm, and I recommend it without any reservation.

Our response:

We thank the Reviewer for summarizing our work, and for his/her very positive assessment of our results, and recommending it for publication in Nature Communications.

Comment 2

I am not familiar with the notion of limit points? What do these correspond to in a nonlinear physical system? Do these points exist in linear systems?

Our response:

This is a good question. We apologize for not having specified this before. Here, LP –as computed with bifurcation continuation packages– stands for limit point or *fold*, where two steady state solutions annihilate each other. Therefore, they only occur in nonlinear systems. In bifurcation theory, a typical fold is a saddle-node bifurcation, where “node” stands for stable equilibrium. In our case the folds are not saddle-nodes because solutions in a neighborhood of the EPs are always unstable, and therefore we used the generic term *fold*. We have now clarified this definition in the caption of Fig. 3.

Comment 3:

Can the authors elaborate more on the relation between EPs and Hopf bifurcations in nonlinear systems?

Our response:

EPs are points where the underlying Hamiltonian matrix is defective, i.e. it has to do with coalescing steady states. On the other hand, Hopf bifurcation refers to the situation where a steady state becomes unstable (the real parts of two eigenvalues of the Jacobian become

positive), giving rise to a limit cycle. So, in general, there is no reason to expect a certain relation between these two points.

Comment 4:

Could the authors comment on if their findings will change in the case of multimode lasers?

Our response:

This is indeed an excellent question. To some extent, our system, when considered as whole is a multimode system as it supports two linear modes. However, in the current case, the two modes interact only through the coupling between the two resonators. The Referee is probably referring to the situation when there more than one mode in the same cavity. In that case, the dynamics will be more complex due direct interaction between the modes via Kerr nonlinear effects and gain saturation nonlinearities. While it is difficult to predict the physics of such complex system without performing simulations and/or experiments, it is not very unreasonable to expect that such a system may exhibit multiple EPs with several transitions between lasing and self-termination states.

We have added the following sentence at the end of the discussion section (page 8, 1st column, 1st. par): “Additionally, it will be interesting to extend our current study to multimode laser systems where the interplay between non-Hermiticity and direct modal interaction may give rise to even more interesting and complex behavior. We investigate these open questions in future works.”

Reviewer #3:

Comment 1

The authors report a study of EP above lasing threshold in a system of coupled micro-resonators. The authors show that detuning of one of the micro-resonators must be designed in to counteract the frequency shift induced by the carriers in the semiconductors (InGaAsP) to achieve EP in a lasing stage. The authors show spectral plots (Fig 3) as the pump power distribution between the two micro-resonators are varied. Of note, the lasing EPs shown in this work are unstable.

Our response:

We thank the Referee for summarizing our work.

Comment 2

While I commend the authors on their effort, the last point presents itself as a serious drawback to me. I cannot think of a use for such a laser presented in this work. I do not think that rich dynamics alone can justify its significance. Unless the authors can persuade otherwise, I regret I cannot support the publication of this work.

Our response:

We respectfully disagree with the Referee's assessment of our work. Although we understand that it would have been nice to reach stable EP laser modes with two coupled cavities, however, this is not possible in two coupled semiconductor microcavities. We believe that it is of fundamental importance to prove it, and to learn what are its consequences, rather than taking this as a drawback. We are shedding light onto something that has been completely overlooked so far: what happens in the proximity to EPs above laser threshold in two coupled lasers. We believe that we addressed this important fundamental question to a large extent in this work. Let us elaborate on two important points:

- 1) As we explained in our manuscript, the goal of this work is to understand the interplay between non-Hermitian and nonlinear effects in laser systems. This crucial point has been completely overlooked in previous works on PT symmetric lasers. Importantly, this lack of understanding to nonlinear non-Hermitian systems poses a serious challenge for engineering laser systems based on non-Hermitian symmetries and for assessing the performance of EP-based sensors in laser platforms. In particular, with respect to the latter problem, there has been a considerable progress in evaluating linear sensors based on EPs (though there is still some debate in the literature). However, nonlinear EP-sensors have been completely overlooked despite the presence of experimental results. The main reason is that these experimental works have relied on linear models to describe the system in order to simplify the analysis. Our work thus provides a first comprehensive effort to understand the nonlinear dynamics of laser systems operating at or close to EPs, including the detailed assessment of their stability features.
- 2) We acknowledge that the referee's point "I do not think that rich dynamics alone can justify its significance..." seem to indicate that the assessment of dynamical behaviors close to EPs were at a very general level in the previous version and needs clarification. First of all, notice that "rich dynamics" is far from being an abstract concept: besides cw laser operation, non-steady states are of fundamental importance as well, also for

applications. To stress this point, in the present version we elaborate more on the predicted non-steady dynamics at the right-hand side of the EPs (Figs. 3 panels d to f). Indeed, the experimental spectra around EPs cannot resolve the actual spectral features and lines become fuzzy. An important example is the frequency combs that are observed in the numerical maps of Figs. 3d, 3e and 3f in the proximity to the EPs, namely at the right-hand side of the EP where no stable steady states exist. The predicted comb frequency is ~ 12 GHz (corresponding to $\Delta\lambda \sim 0.1$ nm) which cannot be resolved by our spectrometer, whose resolution is ~ 200 pm (~ 30 GHz). Such a frequency comb is a consequence of Q-switch self-pulsing. This predicted Q-switch behavior is shown in a new figure [Fig. S4(b), upper panel], and we added the following comment in the text (p. 5, 1st column, 1st par): “These instabilities generate side-bands in the numerical spectra, for instance the observed frequency combs in the simulation panels of Figs. 3(d)-(f), at the right-hand side of EPs where no stable states exist. The predicted comb frequency is $\Delta\nu_{\text{comb}} \sim 12$ GHz, which is governed by the slow electronic lifetime of the system and it can be related to Q-switch like pulsation in coupled nanolasers [51]. Such spectral features cannot be experimentally resolved by in our setup since: i) the spectrometer resolution is ~ 30 GHz, and ii) the pulsed pumping scheme (100-ps pulse duration) is not well adapted to observe such a 12 GHz pulsation.”. We have also added the following comment in the discussions: “On the other hand, some of those instabilities in the proximity to EPs have been shown to generate Q-switch self-pulsing, which open interesting prospects for the realization of non-steady laser sources such as self-pulsing nanolaser devices and nanophotonic frequency combs.”. Finally, we modified the last sentence in the abstract as follows: “This work unveils the unstable nature of EPs above laser threshold in two coupled semiconductor lasers, enabling interesting prospects for the realization of nonsteady laser sources such as self-pulsing nanolaser devices and nanophotonic frequency combs”.

In short, we hope that we have now clarified the importance of our work in a twofold sense: 1) we address the fundamental question of EPs above a laser threshold in a semiconductor dimer, as an interplay between non-Hermiticity and nonlinearity, and 2) we show that the proximity to EPs above laser threshold in coupled lasers open interesting prospects for the realization of non-steady laser sources such as self-pulsing nanolaser devices and nanophotonic frequency combs.

REVIEWERS' COMMENTS

Reviewer #1 (Remarks to the Author):

Review of revised manuscript NCOMMS-23-08002A

manuscript "Tracking exceptional points above the lasing threshold"

by Dr El-Ganainy and colleagues that has been resubmitted to Nature Communications.

The first referee (myself) asked a huge number of questions about clarifications here and there. And these were answered very satisfactorily. The authors clarified small issues here and there, making this excellent work more understandable for general readers. The initial submission was excellent, and the revised version is outstanding. This work is deserving to be published in this journal, especially in this greatly improved version.

The second referee was very enthusiastic and recommended publication. Like me, that referee asked questions to clarify small issues here and there. And the authors provided a very detailed and clear set of replies. Thus, their excellent work became even better and clearer to readers.

The third referee asked far less questions, and was concerned about applications. That objection is not applicable here. This is not the journal of Applied Physics, nor an IEEE journal, where applications are paramount. This is a journal about science. And most of the science published there is not applied (not now, but might become applied in one or two decades in the future, and only a small subset of it). Hundreds of authors, including my team, have published many papers in Science, Nature, Nature Physics, and Nature Comm, and these do not have any direct applications. At least not at the time of publication. Some led to applications a decade or two or three after publication, while others not. These are not IEEE journals, and are not the J of Applied Physics. Thus, I do not understand the surprising logic used by the last referee.

Reviewer #3 (Remarks to the Author):

I thank the authors for the response and revision. The authors have clarified the significance of their work and thoroughly discussed the role of instability. I support the publication of this manuscript.

Response letter
Tracking exceptional points above the lasing threshold

Reviewer #1:

Overall Comment

The first referee (myself) asked a huge number of questions about clarifications here and there. And these were answered very satisfactorily. The authors clarified small issues here and there, making this excellent work more understandable for general readers. The initial submission was excellent, and the revised version is outstanding. This work is deserving to be published in this journal, especially in this greatly improved version.

The second referee was very enthusiastic and recommended publication. Like me, that referee asked questions to clarify small issues here and there. And the authors provided a very detailed and clear set of replies. Thus, their excellent work became even better and clearer to readers.

The third referee asked far less questions, and was concerned about applications. That objection is not applicable here. This is not the journal of Applied Physics, nor an IEEE journal, where applications are paramount. This is a journal about science. And most of the science published there is not applied (not now, but might become applied in one or two decades in the future, and only a small subset of it). Hundreds of authors, including my team, have published many papers in Science, Nature, Nature Physics, and Nature Comm, and these do not have any direct applications. At least not at the time of publication. Some led to applications a decade or two or three after publication, while others not. These are not IEEE journals, and are not the J of Applied Physics. Thus, I do not understand the surprising logic used by the last referee.

Our response:

We thank again the reviewer for the very positive assessment of our work. We are glad that we could answer all the reviewers' comments and questions.

Reviewer #3:

Overall Comment

I thank the authors for the response and revision. The authors have clarified the significance of their work and thoroughly discussed the role of instability. I support the publication of this manuscript.

Our response:

We thank the reviewer for his/her positive comments. We are glad that we have addressed the reviewer's concern.